# Design and Analysis Considering Magnet Usage of Permanent Magnet Synchronous Generator Using Analytical Method

Ji-Hun Lee [1], Hoon-Ki Lee [1], Young-Geun Lee [1], Jeong-In Lee [1], Seong-Tae Jo [1], Kyong-Hwan Kim [2], Ji-Yong Park [2] and Jang-Young Choi [1],*

[1] Department of Electrical Engineering, Chungnam National University, Daejeon 34134, Korea; ljh@o.cnu.ac.kr (J.-H.L.); lhk1109@cnu.ac.kr (H.-K.L.); 201502206@o.cnu.ac.kr (Y.-G.L.); lji477@cnu.ac.kr (J.-I.L.); jst4900@o.cnu.ac.kr (S.-T.J.)

[2] Offshore Plant Research Division Korea Research Institute of Ships and Ocean Engineering, Daejeon 34103, Korea; kkim@kriso.re.kr (K.-H.K.); jypark@kriso.re.kr (J.-Y.P.)

* Correspondence: choi_jy@cnu.ac.kr

**Abstract:** In this study, the characteristic analysis of a permanent magnet synchronous generator was performed using the analytical method, and the validity of the analytical method was compared with that of the finite element method (FEM). For the initial design, the rotor size was selected using the torque per rotor volume method, and the stator size was selected according to the saturation of the stator iron core. In addition, fast Fourier transform analysis was performed to determine the appropriate magnet thickness point, and it was confirmed that the open circuit and armature reaction magnetic flux densities were consistent with the FEM analysis results. Based on the analytical method, the generator circuit constants (phase resistance, back EMF, and inductance) were derived to construct an equivalent circuit. By applying the equivalent circuit method to the derived circuit constants, the accuracy of the equivalent circuit method was confirmed by comparing the FEM and experimental results.

**Keywords:** permanent magnet synchronous generator (PMSG); analytical method; finite element method (FEM); magnet usage

## 1. Introduction

Permanent magnet synchronous generators (PMSGs) are increasingly used in various applications owing to their high efficiency and high torque-to-volume ratio [1,2]. Electromagnetic analysis is essential to understand the characteristics of PMSG, and the prediction of the magnetic field distribution is crucial. Currently, researchers are performing electromagnetic analyses using various methods, and many studies have been published. An analytical method for calculating the governing equation using the subdomain method is shown in [3–10], a simplified magnetic equivalent circuit method based on the analytical method is shown in [11,12], and the finite element method (FEM) using numerical analysis is shown in [13–19]. In addition, a hybrid calculation method that combines the two methods was used in [20,21]. Currently, FEM with high accuracy is being used frequently, considering the increase in the saturation effect [22]. However, the model needs to be designed directly into a software program, and transient analysis has a long computation time. The analytical method using the space harmonic method is complicated because the governing equation must be directly solved using Maxwell's equation [23,24]. As a disadvantage of this method, the accuracy is lower than the FEM, and the magnetic saturation phenomenon cannot be considered [9,24]. In addition, it is necessary to calculate the undefined coefficient by setting the boundary condition according to the machine shape. However, this method has an advantage over FEM in cases involving various design parameters. Moreover, an important part of the initial design is the analysis time, which is only a few seconds using the analytical method [9].

In this study, for the initial design of the PMSG, the rotor size and axial length were selected using the torque per rotor volume (TRV) method for the initial design of the PMSG, and the stator size was selected according to the design requirements. When designing a machine, the permanent magnet used can be said to be an important factor influencing the performance of the machine. However, if the magnet usage increases, it causes the price to rise. Therefore, in order to reduce magnet usage, harmonic analysis was performed through fast Fourier transform (FFT) according to magnet thickness. As a result of the analysis, the magnetic flux density began to saturate at a magnet thickness of 5 mm. By applying the selected magnet thickness to the initial design, unnecessary magnet usage could be reduced. The magnetic field characteristics of the open circuit and armature reactions were calculated using the subdomain method to elucidate on the design parameters. The calculated results were compared with the FEM analysis results, and circuit constants such as phase resistance, back EMF, and inductance were derived to construct an equivalent circuit of the generator. The equivalent circuit method was applied using the derived circuit constants, and the FEM and experimental results were compared. The validity of the analytical method was proved by showing that the calculation results were consistent. In this study, all FEMs compared with analytical methods were analyzed using a commercial software, ANSYS Maxwell.

## 2. Initial Design

### 2.1. Process of Initial Design

In this study, the initial design of the PMSG was performed according to the design flowchart shown in Figure 1. First, after checking the design requirements in Table 1, the rotor size was selected using the TRV method, as shown in Figure 2. Thereafter, the structure and number of turns of the stator were selected according to the design requirements, and circuit constants such as inductance were derived by analyzing the no-load magnetic field characteristics.

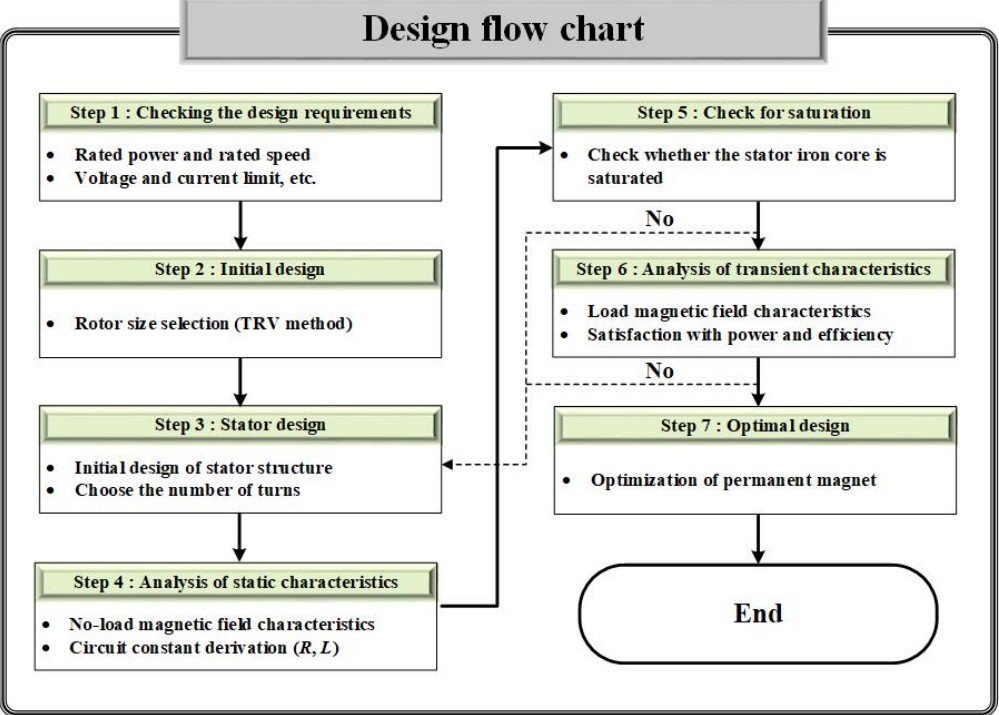

**Figure 1.** PMSG design flow chart.

**Table 1.** Design requirements.

| Parameter | Value | Unit |
|---|---|---|
| Rated power | 500 | W |
| Rated speed | 300 | rpm |
| Permanent magnet | N35SH | - |
| Efficiency | 90 | % |
| Residual magnetic flux density | 1.21 | T |
| Current density limit | 7 | A/mm$^2$ |

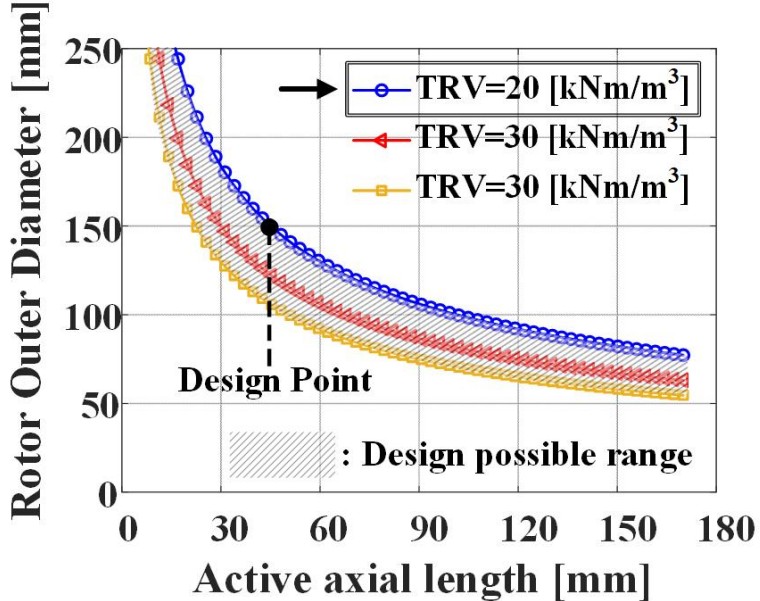

**Figure 2.** TRV curve.

At this time, if the saturation of the iron core of the stator is satisfied, the load magnetic field characteristic analysis may be performed, but if not satisfied, the stator design should be performed again. Similarly, when the load magnetic field characteristic analysis is performed, the stator design must be repeated if the requirements are not satisfied. Conversely, the performance of the device can be improved by optimizing the permanent magnet if the requirements are satisfied.

*2.2. Size Selection*

The TRV method was used to select the initial design model of the PMSG. Here, TRV is the torque that can be used in a given rotor volume per unit rotor volume; it can be observed as an empirical value by the designer and varies depending on the material of the permanent magnet [25]. The TRV method can be calculated according to Equation (1) and can be represented by the TRV curve shown in Figure 2. In addition, because the material of the permanent magnet used is the neodymium series, the TRV value was selected as 20 kNm. Finally, the rotor size and axial length were selected as 150 and 45 mm, respectively.

$$TRV = \frac{T_{out}}{\frac{\pi}{4} D_{ro}^2 L_{stk}} \tag{1}$$

where $D_{ro}$ is the outer diameter of the rotor, $L_{stk}$ is the axial length, and $T_{out}$ is the output torque. The stator size was selected as 250 mm according to the design requirements listed in Table 1, and a surface PMSG with a small torque ripple and easy control was selected. After the voltage applied to the inverter is determined, the number of turns can be calculated by determining the value of the back EMF. The back EMF induced in the state in

which the armature winding is an open circuit can be calculated according to Equation (2), and the number of turns per slot can be calculated according to Equation (3) [25].

$$E_{max} = N_p N_t k_d k_p B_r L_{stk} \gamma \omega_n \tag{2}$$

$$N_s = \frac{E_r}{N_p N_{spp} k_d k_p B_r L_{stk} \gamma \omega_m} \tag{3}$$

where $N_p$ is the number of poles, $k_d$ is the distribution factor, $k_p$ is the short-pitch factor, $B_r$ is the air-gap flux density, $N_t$ is the number of turns per phase, and $\omega_n$ is the rotor angular velocity. Compared to distributed winding, concentrated winding produces less heat and fractional pole/slot combination is possible; therefore, it is advantageous to improve the cogging torque and the back EMF [26,27]. Therefore, concentrated winding was applied. In addition, concentrated winding is easy to manufacture with small resistance and copper loss as the length of the end coil is short. Figure 2 shows the final selected analysis model and the manufactured prototype.

## 3. Magnetic Field Characteristics Analysis Using Analytical Methods

### 3.1. Analytical Method

The analytical method is used to analyze the magnetic field distribution by calculating the magnetic vector potential from Maxwell's equation. During calculation, the domain was divided using the subdomain method with several assumptions. In each domain, the governing equations were calculated based on Maxwell's equations, and the undefined coefficients were derived using the appropriate boundary conditions. Figure 3 shows the general calculation process for calculating the magnetic field distribution using the analytical method.

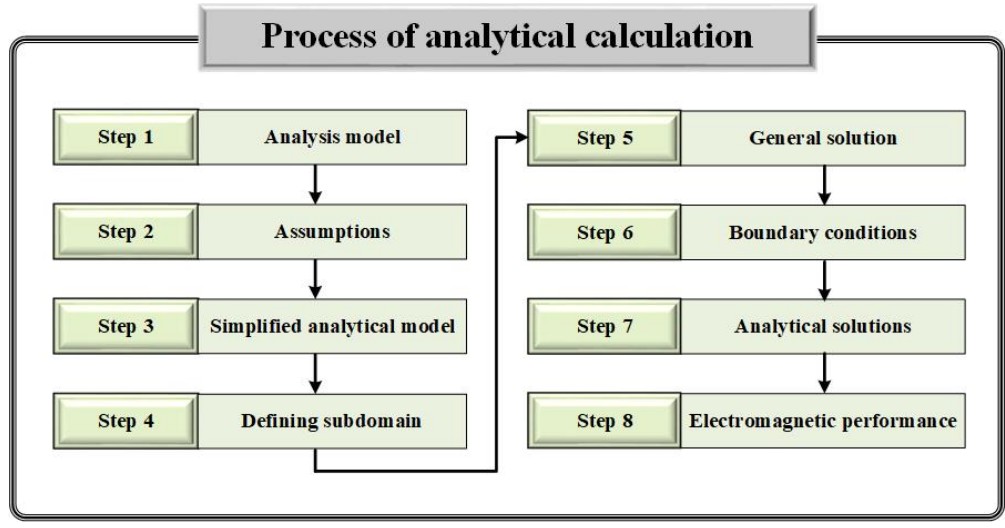

**Figure 3.** Process of analytical calculation.

### 3.2. Assumptions for Analysis

Figure 4 shows a simplified model of the analysis of the magnetic field characteristic. In the initial design of the machine, it is important to quickly secure the tendency of the machine characteristics. This is because designers conduct design and characteristic analysis several times to satisfy design requirements. Using the analytical method that takes only a few seconds, the tendency of machine characteristics can be quickly obtained. In addition, even if teeth were not included, a result with high accuracy was derived from the analysis result. Therefore, the simplified model was analyzed as a slotless model. The analysis region consisted of an air-gap region (I) and a permanent magnet region (II) for the analysis of the magnetic field characteristic. For the analysis of the magnetic

field distribution characteristics of the open circuit and armature reactions, the magnetic permeability of the stator and rotor iron cores could be assumed to be infinite. In addition, the relative permeability of the permanent magnet material and the air gap was assumed to be 1, which is the same as that of air. Because the relative permeability of neodymium-based permanent magnets is close to 1, there was no significant difference in the analysis results. Therefore, for the convenience of calculation, the relative permeability of the permanent magnet was determined to be 1. When performing the magnetic field distribution analysis of the armature reaction, it was considered that a current sheet was applied to the stator core. The main parameters of the analytical model presented in this study are shown in Table 2.

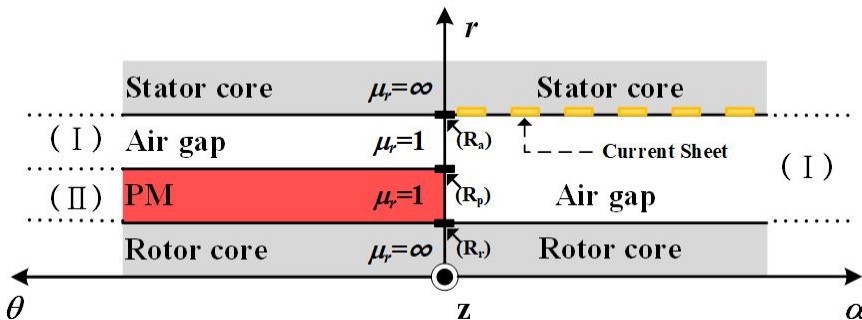

**Figure 4.** Simplified analytical model.

**Table 2.** Parameters of simplified analytical model.

| Parameter | Designation | Unit |
|-----------|-------------|------|
| $R_a$ | Inner radius of stator | mm |
| $R_p$ | Outer radius of PMs | mm |
| $R_r$ | Inner radius of PMs | mm |
| $\alpha_p$ | Pole-arc ratio | - |
| $s_o$ | Width of slot opening | mm |
| $l_{stk}$ | Axial length | mm |

*3.3. Magnetization Modeling and Current Modeling*

Because the magnetization direction of the permanent magnet of this model is parallel magnetization, it is expressed as Equation (4) by a Fourier series.

$$\mathbf{M} = \sum_{n=1,odd}^{\infty} \left\{ M_{rn}\cos(q\theta)\mathbf{i_r} + M_{\theta n}\sin(q\theta)\mathbf{i_\theta} \right\} \tag{4}$$

$$M_{rn} = \begin{vmatrix} M_0\alpha_p[K_{mo}(q+1) + K_{mo}(q-1)] & n = ip, i = 1, 3, 5, \dots \\ M_0\alpha_p[K_{mo}(q+1)] & n = ip = 1 \\ 0 & otherwise \end{vmatrix} \tag{5}$$

$$M_{\theta n} = \begin{vmatrix} M_0\alpha_p[K_{mo}(q+1) - K_{mo}(q-1)] & n = ip, i = 1, 3, 5, \dots \\ M_0\alpha_p[K_{mo}(q+1)] & n = ip = 1 \\ 0 & otherwise \end{vmatrix} \tag{6}$$

where $M_{rn}$ and $M_{\theta n}$ are the Fourier coefficients in the $r$ and $\theta$ directions, respectively; $q$ is expressed as $n$ ($n$th harmonic) $\times p$ (pole pair); and $i_r$ and $i_\theta$ indicate unit vectors in the $r$ and $\theta$ directions, respectively. In addition, $M_0 = B_r/\mu_0$ indicates that $\alpha_p$, $n$, $B_r$, and $\mu_0$ are the pole-arc ratio, harmonic order, residual magnetic flux density of the permanent magnet,

and magnetic permeability of the vacuum, respectively. The coefficient $K_{mo}(\bullet)$ is given according to Equation (7).

$$K_{mo}(\bullet) = \frac{\sin\left(\bullet \cdot \alpha_p \cdot \frac{\pi}{2}\right)}{\left(\bullet \cdot \alpha_p \cdot \frac{\pi}{2}\right)} \tag{7}$$

A magnetic field is generated by an armature reaction that occurs when the current flows through the windings of the stator. To derive this, current modeling was performed according to Equations (8)–(10). The *n*-order Fourier coefficient for the current density distribution is expressed by Equation (11).

$$J_a = \sum_{n=1,odd}^{\infty} I_n i_a \cos(q\theta) \tag{8}$$

$$J_b = \sum_{n=1,odd}^{\infty} I_n i_b \cos\left\{q(\theta - \frac{2}{3}\frac{\pi}{p})\right\} \tag{9}$$

$$J_c = \sum_{n=1,odd}^{\infty} I_n i_c \cos\left\{q(\theta + \frac{2}{3}\frac{\pi}{p})\right\} \tag{10}$$

$$I_n = \frac{2N_{ppt}}{ns_o\pi}\left[\sin\left\{q\frac{s_o}{2R_a}\right\} + \sin\left\{q\left(\frac{\pi}{p} - \frac{s_o}{2R_a}\right)\right\}\right] \tag{11}$$

where $i_a$, $i_b$, and $i_c$ are the maximum current values of phases *a*, *b*, and *c*, respectively, and the phases have a phase difference of 120°. $N_{ppt}$, $S_o$, and $R_s$ represent the number of turns per pole, the length of the slot opening, and the inner radius of the stator, respectively. Therefore, three-phase current modeling can be expressed according to Equation (12).

$$\begin{aligned} J &= J_a + J_b + J_c \\ &= \sum_{n=1,odd}^{\infty} I_n\left[i_a\cos(q\theta) + i_b\cos\left\{q(\theta - \frac{2}{3}\frac{\pi}{p})\right\} + i_c\cos\left\{q(\theta + \frac{2}{3}\frac{\pi}{p})\right\}\right] \end{aligned} \tag{12}$$

### 3.4. Selection of Appropriate Magnet Usage Point

The performance of the device improves as the usage of the magnet increases, but the cost increases; therefore, appropriate magnet usage is required. Studies on replacing rare earth permanent magnets with ferrite magnets to reduce magnet usage are presented in [28–30] and deal with the reduction in magnet usage through pole-arc change. In this study, magnet usage is reduced only by changing the thickness of the magnet without changing the design parameters. Figure 5 shows the FFT analysis of the 1st harmonic to find the appropriate magnet usage point before performing the open circuit and armature reaction magnetic field characteristics analysis. First, using the analytical method, the thickness of the magnet was analyzed from 1 mm to 10 mm at intervals of 1 mm. As a result, the magnetic flux density began to saturate at a magnet thickness of 5 mm, and it could be observed that the change in the magnetic flux density converged as the thickness increased. Therefore, in this study, a magnet thickness of 5 mm, which is the point with the largest magnet usage versus magnetic flux density, was selected as the appropriate point.

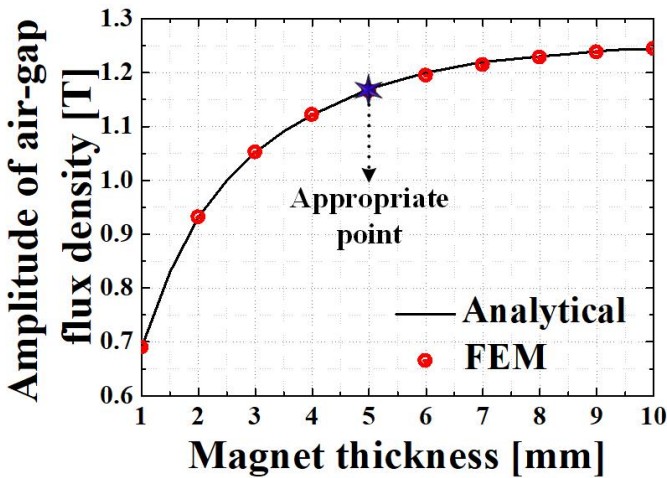

**Figure 5.** FFT analysis according to magnet thickness.

### 3.5. Open Circuit Magnetic Field Characteristics

The expression for the magnetic flux density is **B** = $\mu_0$ (**H** + **M**), and it becomes **H** = 0 when considering the magnetic field characteristics of the open circuit because there is no displacement current component. It becomes an open circuit when there is no current component, and the magnetic flux density can be obtained by the permanent magnet. Thereafter, by taking $\nabla$ on both sides and applying the vector identity formula, $\nabla \times (\nabla \times \mathbf{A}) = \nabla(\nabla \cdot \mathbf{A}) - \nabla^2\mathbf{A}$, Equation (13) can be obtained.

$$\nabla^2\mathbf{A} = -\mu_0(\nabla \times \mathbf{M}) \tag{13}$$

Using Equation (13), the air-gap region (I) does not have a magnetization component; therefore, it can be expressed using the Laplace equation, and the permanent magnet region (II) has a magnetization component and can be expressed using the Poisson equation. Equations (14) and (15) show the governing equations for the two domains.

$$\nabla^2\mathbf{A}^\mathrm{I} = 0 \tag{14}$$

$$\nabla^2\mathbf{A}^\mathrm{II} = -\mu_0(\nabla \times \mathbf{M}) \tag{15}$$

In a cylindrical machine, the magnetic vector potential, **A**, acts in the z-direction; therefore, it can be expressed as **A** = $A(r)\sin(q\theta)\mathbf{i_z}$. In addition, using $\nabla \times \mathbf{A} = \mathbf{B}$, the definition of the magnetic vector potential, is expressed according to Equations (16)–(19), and the boundary conditions for deriving the undefined coefficient are listed in Table 3. Figure 6 shows a comparison of the open circuit magnetic field characteristics, and it was confirmed that the analysis results were consistent. At this time, the analysis was performed without applying a current in the FEM as well.

$$\mathbf{B}_{rn}^\mathrm{I} = \sum_{n=1,3,5,\dots}^{\infty} \frac{q}{r}\left[C_n^\mathrm{I}r^q + D_n^\mathrm{I}r^{-q}\right]\cos(q\theta)\mathbf{i}_r \tag{16}$$

$$\mathbf{B}_{rn}^\mathrm{II} = \sum_{n=1,3,5,\dots}^{\infty} \frac{q}{r}\left[C_n^\mathrm{II}r^q + D_n^\mathrm{II}r^{-q} - \frac{\mu_0 q r M_n}{1-q^2}\right]\cos(q\theta)\mathbf{i}_r \tag{17}$$

$$\mathbf{B}_{\theta n}^\mathrm{I} = \sum_{n=1,3,5,\dots}^{\infty} q\left[C_n^\mathrm{I}r^{q-1} - D_n^\mathrm{I}r^{-(q+1)}\right]\sin(q\theta)\mathbf{i}_\theta \tag{18}$$

$$\mathbf{B}_{\theta n}^\mathrm{II} = \sum_{n=1,3,5,\dots}^{\infty} -q\left[C_n^\mathrm{II}r^{q-1} - D_n^\mathrm{II}r^{-(q+1)} - \frac{\mu_0 M_n}{1-q^2}\right]\sin(q\theta)\mathbf{i}_\theta \tag{19}$$

**Table 3.** Boundary conditions for deriving undefined coefficients.

| Open Circuit | | Armature Reaction | |
|---|---|---|---|
| $(r = R_a)$ | $\mathbf{B}_{\theta n}^{\mathrm{I}} = 0$ | $(r = R_a)$ | $\mathbf{B}_{\theta n}^{\mathrm{I}} = -\mu_0 J_a$ |
| $(r = R_p)$ | $\mathbf{B}_{rn}^{\mathrm{II}} = \mathbf{B}_{rn}^{\mathrm{I}}$ | | |
| $(r = R_p)$ | $\mathbf{B}_{rn}^{\mathrm{II}} - \mathbf{B}_{rn}^{\mathrm{I}} = \mu_0 M_{\theta n}$ | $(r = R_r)$ | $\mathbf{B}_{\theta n}^{\mathrm{I}} = 0$ |
| $(r = R_r)$ | $\mathbf{B}_{\theta n}^{\mathrm{II}} = \mu_0 M_{\theta n}$ | | |

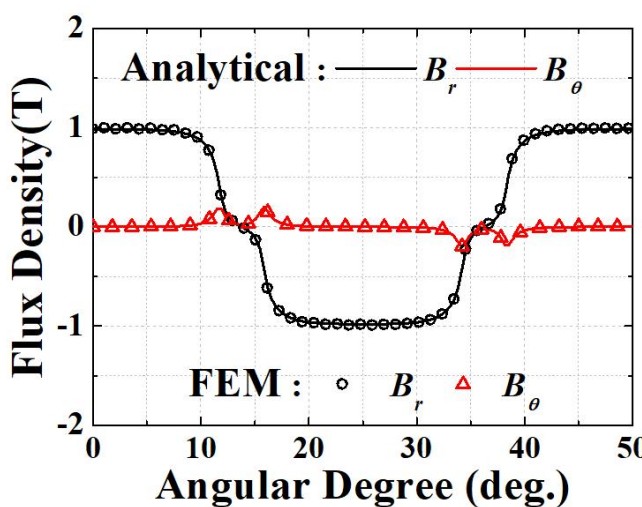

**Figure 6.** Comparison of open circuit magnetic field characteristics.

*3.6. Magnetic Field Characteristics of Armature Reaction*

The magnetic field of the armature reaction is similar to the process of deriving the magnetic field of the open circuit, but it is assumed that there is no permanent magnet, and the current sheet is applied to make $M = 0$. In addition, because it is assumed that the permeability of the iron core is infinite, it becomes $\mathbf{H} = 0$ when it is applied to $\mathbf{B} = \mu\mathbf{H}$; thus, $\nabla^2 \mathbf{A} = 0$ is derived. Here, if $\nabla \times \mathbf{A} = \mathbf{B}$, which is the definition of the magnetic vector potential, is applied, a magnetic flux density equation such as Equations (20) and (21) can be derived.

$$\mathbf{B}_{rn}^{\mathrm{I}} = \sum_{n=1,3,5,\ldots}^{\infty} -\frac{jq}{r} \left[ C_n^{\mathrm{I}} r^q + D_n^{\mathrm{I}} r^{-q} \right] e^{-jq\theta} i_r \tag{20}$$

$$\mathbf{B}_{\theta n}^{\mathrm{I}} = \sum_{n=1,3,5,\ldots}^{\infty} -q \left[ C_n^{\mathrm{I}} r^{q-1} + D_n^{\mathrm{I}} r^{-(q+1)} \right] e^{-jq\theta} i_\theta \tag{21}$$

In addition, the undefined coefficient was derived using the boundary conditions listed in Table 3 for the magnetic field characteristics of the armature reaction. Figure 7 shows the comparison of magnetic field characteristics by the armature reaction. For accurate analysis results, the characteristic analysis was performed with the same initial position as in the method used in this study and the FEM.

Figure 8 is comparison of magnetic field characteristics considering both open circuit and armature reaction. When compared with Figure 6, it can be seen that there is no significant difference. Because, when analyzing the armature reaction magnetic field characteristics, the equation of $i_a = -i_b - i_c$ was used. That is, the calculation was performed with $i_a = 1$ [A], $i_b = -0.5$ [A], and $i_c = -0.5$ [A].

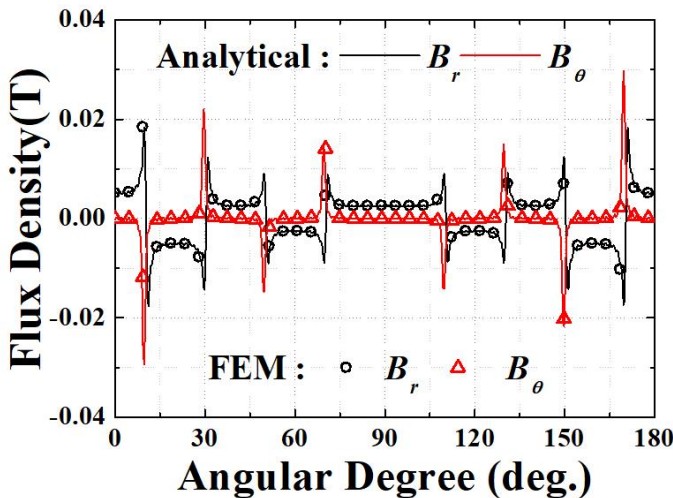

**Figure 7.** Comparison of armature reaction magnetic field characteristics.

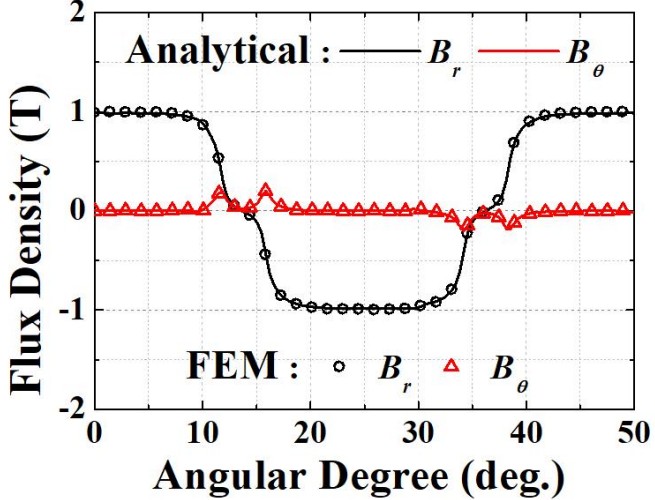

**Figure 8.** Comparison of magnetic field characteristics considering both open circuit and armature reaction.

## 4. Equivalent Circuit Method (ECM)

### 4.1. Circuit Constant Derivation

#### 4.1.1. Phase Resistance

The circuit constant must be derived to construct the equivalent circuit required for the analysis of the output characteristics of the PMSG. Electromagnetic performance can be analyzed by coupling the proposed analytical model with the ECM. The phase resistance was calculated using Equation (22).

$$R_{ph} = \rho_c \frac{l_c}{A_c N_{sn}} \tag{22}$$

where $A_c$ and $N_{sn}$ represent the cross-sectional area of the conductor and the number of strands, respectively. Because $\rho_c$ is the resistance of copper and it varies with temperature, it can be calculated according to Equation (23).

$$\rho_c = \rho_0\{1 + \alpha_r(T - T_0)\} \tag{23}$$

where $\rho_0$ represents the resistivity at room temperature ($T_0$) and it has a value of $\rho_0 = 1.724 \times 10^{-8}$ [$\Omega \cdot$m] for general copper. Finally, the total length, $l_c$, of the coil is calculated according to Equation (24).

$$l_c = N_{pc} N_{ct} (2 \times L_{stk} + 2\pi r_e) \tag{24}$$

where $N_{pc}$, $N_{ct}$, and $L_{stk}$ denote the number of coils per phase, number of turns per coil, and axial length, respectively. In addition, because there are two end-turn parts, it can be expressed as $2\pi r_e$. Figure 9 shows a conceptual diagram to elucidate the phase-resistance calculation.

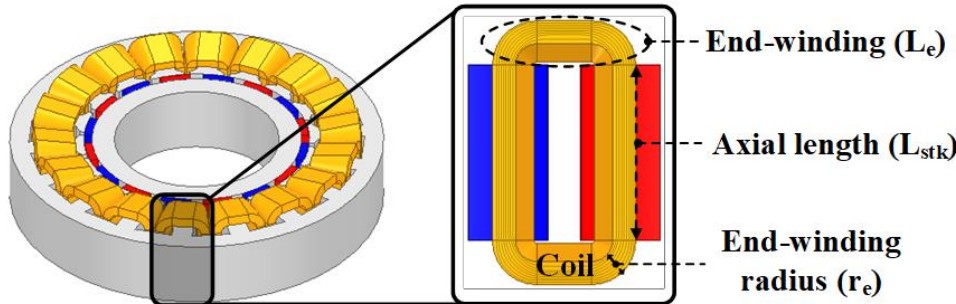

**Figure 9.** Conceptual diagram for calculating phase resistance.

### 4.1.2. Inductance

The inductance that can be derived from the magnetic field of the armature reaction is crucial as a circuit constant that affects the power factor and efficiency. Inductance is the amount of flux linkage per unit current and it is expressed as the sum of the self-inductance and mutual inductance [27]. In this study, the magnitude of the inductance occurring at the end-turn part is very small; therefore, it is not considered, and the self-inductance can be calculated according to Equation (25).

$$L_{self} = \frac{\lambda_a}{i_a} \tag{25}$$

where $i_a$ is the maximum current of phase a, and because it is a three-phase balanced load, a phase difference of 120° occurs. Therefore, $i_a = -i_b - i_c$, and the mutual inductance can be expressed as $M = -0.5L_{self}$. The flux linkage by phase A can be calculated according to Equation (26), where $L_s$ is the synchronous inductance.

$$\lambda_a = L_{self} i_a + M i_b + M i_c = 1.5 L_{self} i_a = L_s i_a \tag{26}$$

### 4.1.3. Back EMF

Back EMF is the voltage generated in the coil by the flux linkage generated while the rotor rotates, and it can be derived using the magnetic field characteristics of the permanent magnet. The back EMF of the coil generated in one phase is expressed according to Equation (27) using Faraday's law.

$$E_{emf} = -\frac{d\lambda}{dt} \tag{27}$$

When the flux linkage generated by the permanent magnet is substituted in Equation (27), the back EMF generated in one phase can be expressed according to Equation (28). In addition, the back EMF constant can be calculated according to Equation (29).

$$E_{emf} = 2\omega_r p N_{ppt} R_s l_z \sum_{n=1,odd} B_n \sin\left(q\frac{\alpha_y}{2}\right) \sin(q\omega_r t) \tag{28}$$

$$K_e = \max\left(\frac{E_{emf}}{\omega_r}\right) \tag{29}$$

### 4.2. Composition of Equivalent Circuit

To analyze the output characteristics of the PMSG, an equivalent circuit composition is essential. Figure 10 shows the equivalent circuit composed of the circuit constants derived in Section 4.1. Equations (30)–(33) are formulas for deriving the terminal voltage per phase ($V_t$), current per phase ($I_{ph}$), induced electromotive force per phase ($E_{ph}$), and rated power ($P_0$).

$$V_t = I_{ph}R_{load} = E_{ph} \times \frac{R_{load}}{\sqrt{(R_{ph} + R_{load})^2 + X_s^2}} \tag{30}$$

$$I_{ph} = \frac{V_t}{R_{load}} = \frac{E_{ph}}{\sqrt{(R_{ph} + R_{load})^2 + X_s^2}} \tag{31}$$

$$E_{ph} = V_t + I_{ph}R_{ph} + jI_{ph}X_s \tag{32}$$

$$P_0 = V_{ta}I_a + V_{tb}I_b + V_{tc}I_c \tag{33}$$

where $R_{load}$ is the load resistance and $X_s$ is the synchronous reactance. $X_s$ can be derived as $X_s = 2\pi f L_s$, and it represents the electrical frequency. In addition, the terminal voltage and current per phase were calculated as root mean square values.

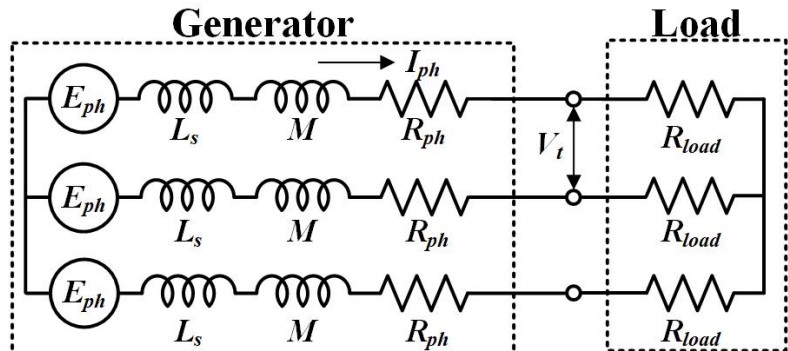

**Figure 10.** Equivalent circuit of PMSG.

### 4.3. Comparison of Generating Characteristics Analysis Results

Figure 11 shows the FEM analysis model and the actual model. Table 4 lists the specifications of the manufactured PMSG based on the design requirements. Figure 12 shows the device for the experimental verification of the method used in this study. The experimental setup consisted of a PMSG, torque sensor, driving motor, inverter, and power analyzer to measure the power generation characteristics. In addition, the rated power was over 500 W and the current density was 3.02 A/mm$^2$; therefore, the design requirements were satisfied.

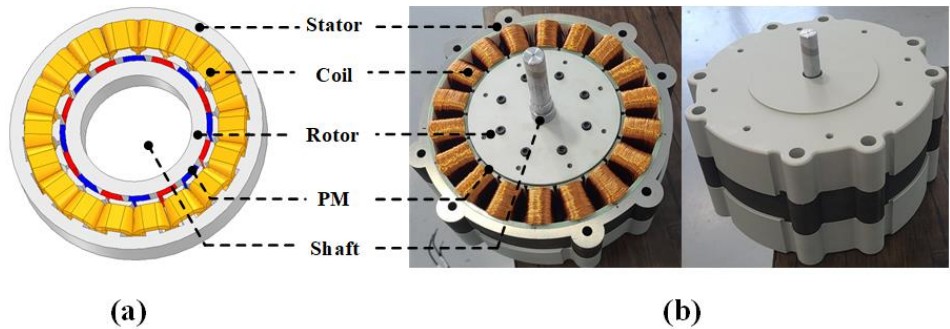

**Figure 11.** Analysis model: (**a**) FEM analysis model and (**b**) actual model.

**Table 4.** Specifications of the actual PMSG.

| Parameter | Value | Unit |
|---|---|---|
| Pole/slot | 16/18 | - |
| Stator diameter | 250 | mm |
| Rotor diameter | 150 | mm |
| Axial length | 45 | mm |
| Magnet thickness | 5 | mm |
| Winding | Concentrated | - |
| Pole-arc ratio | 0.82 | - |
| Air gap length | 1 | mm |
| Coil diameter | 0.511 | mm |
| Number of strands | 17 | - |
| Number of turns | 25 | - |

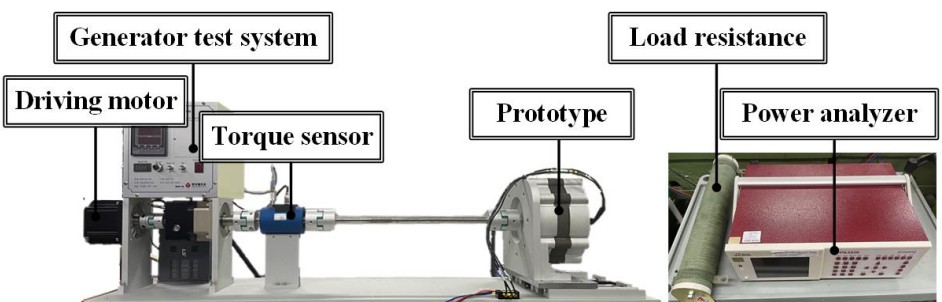

**Figure 12.** PMSG setup for experimental verification.

Figure 13 shows the comparison of back EMF results of analytical methods, FEM (slotless model), FEM (slot model), and actual experiments. There is no significant difference between the slotless model applied in this study and the slot model. Table 5 summarizes the comparison between the phase resistance and back EMF constant, and the phase resistance was analyzed with the same value. The back EMF constant shows similar values, although there is a slight error. Figure 13 shows the current, voltage, power, and efficiency curves according to the change in the load resistance at the rated speed. We wanted more experimental results, but we could not add more due to the lack of number and variety of load resistances. However, as shown in Figure 14, the accuracy of the characteristic analysis results according to the load resistance is excellent. Therefore, it is considered to have high accuracy of the experimental results at another load resistance point. Although the experimental value of the efficiency is excluded from Figure 14d, it is considered that the efficiency is consistent because the comparison of other characteristic values is consistent.

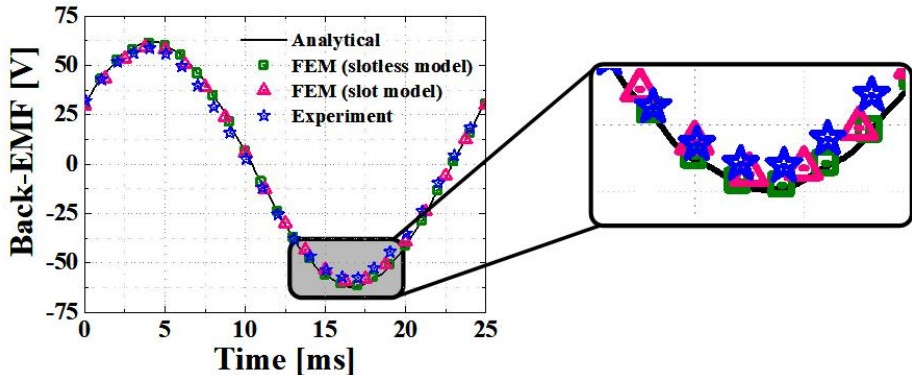

**Figure 13.** Comparison of back EMF analysis.

**Table 5.** Comparison of phase resistance and back EMF constant.

|  | Analytical | FEM | Experiment |
|---|---|---|---|
| Phase resistance | 0.16 | 0.16 | 0.16 |
| Back EMF constant | 1.38 | 1.33 | 1.32 |

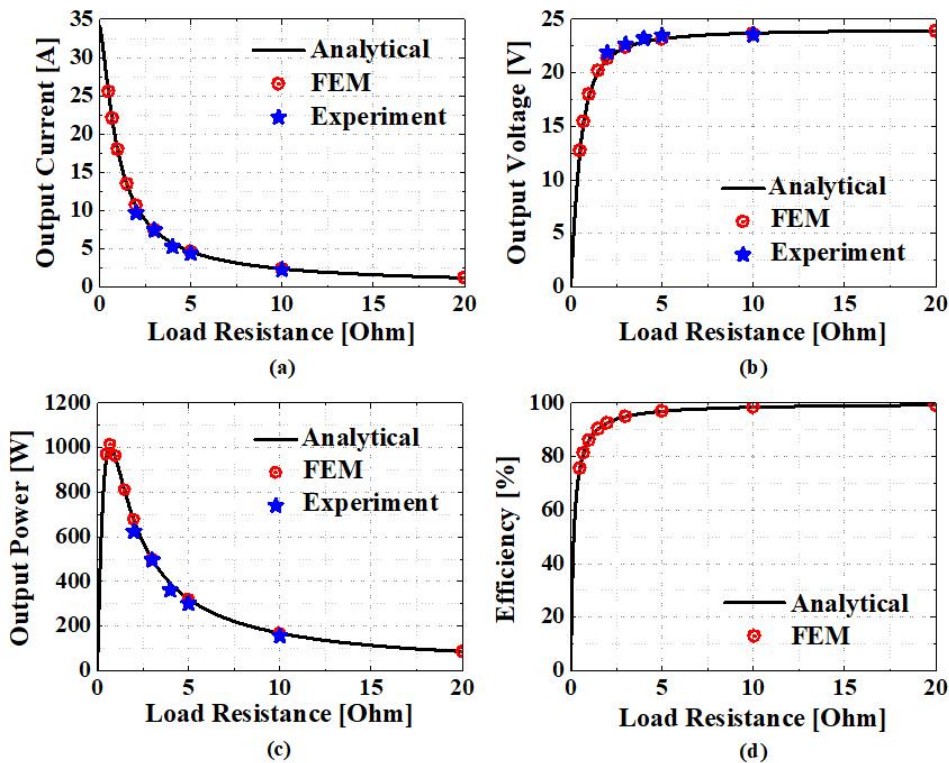

**Figure 14.** Results of characteristic analysis: (**a**) current curve, (**b**) voltage curve, (**c**) power curve, and (**d**) efficiency curve according to load resistance.

## 5. Conclusions

In this study, the size of the rotor was selected using the TRV method to perform the initial design of the PMSG, and the size of the stator was selected considering the saturation of the stator core. After performing the initial design, the magnetic field characteristics of the no-load and load were analyzed. In addition, to use the analytical method, which has the advantage of rapid initial design, the domain was divided using the subdomain method, and several assumptions were made. After performing magnetization modeling

and current modeling, FFT analysis was performed by changing only the thickness of the magnet using an analytical method to reduce the magnet usage. After verifying the analysis results through FEM, the point with the largest magnet usage versus magnetic flux density was derived. In addition, boundary conditions were applied to calculate the undefined coefficients of the open circuit and armature reaction. It was compared with the magnetic flux density of the FEM, and it was consistent. Circuit constants (phase resistance, inductance, and back EMF) were derived using the ECM method based on the analytical method. The analysis was performed according to the change in load resistance by the analytical method and compared with FEM. Additionally, the results of the experimental setup for verification were consistent. Therefore, the validity of the proposed method was verified in this study.

**Author Contributions:** J.-Y.C., conceptualization, review, and editing; J.-H.L., original draft preparation and electromagnetic analysis; H.-K.L., experiment and editing; Y.-G.L., experiment and software; J.-I.L., visualization and review; S.-T.J., visualization and software; K.-H.K., funding acquisition and supervision; J.-Y.P., funding acquisition and supervision. All authors have read and agreed to the published version of the manuscript.

**Funding:** This work was supported in part by the Korea Research Institute of Ships and Ocean Engineering through Endowment Project of "Development of Wave Energy Converter Analysis (WECAN) for the Establishment of Integrated Performance and Structural Safety Analytical Tools of Wave Energy Converter" under Grant PES3530, and this research was supported by a grant from the National R&D Project "Development of Wave Energy Converters Applicable to Breakwater and Connected Micro-Grid with Energy Storage System" funded by Ministry of Oceans and Fisheries, Korea (PMS4590).

**Institutional Review Board Statement:** Not applicable.

**Informed Consent Statement:** Not applicable.

**Data Availability Statement:** The data presented in this study are available on request from the corresponding author.

**Conflicts of Interest:** The authors declare no conflict of interest.

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
