# Peer review of "Design and Analysis Considering Magnet Usage of Permanent Magnet Synchronous Generator Using Analytical Method"

_electronics, doi:10.3390/electronics11020205_

Round 1

Reviewer 1 Report

Space after comma, even in cases as  [1,2] etc.

Caption of Figure 2.TRV curve”. In Fig. 2 there are three curves!

The row after equations, has to be left aligned *e.g. Eqs. (1), (3), (6), etc.

Last sentence before title of Section 3: “Figure 2 shows the final selected analysis model and the manufactured prototype.” It must be a mistake, Fig. 2 shows TRV curves.

First sentence of Section 3.1 “The analytical method was calculated based on the Fourier series, which …” A method cannot be calculated. Please change the sentence.

In Section 3.2: “… the magnetic permeability of the stator and rotor iron cores could be infinite …”. Could be assumed to be infinite ….

In Section 3.2: “… the relative permeability of the permanent magnet and the air gap was assumed to be 1 …”. relative permeability of the permanent magnet material and the air gap was assumed

In Section 3.2: “… When performing the magnetic field distribution analysis of the armature reaction, the current sheet was applied to the stator.”  Please change the sentence. You considered that a current sheet was applied to the stator core.

After Eq. (6):”…  are the Fourier coefficients in the r and θ directions, respectively; q is expressed as n (nth harmonic) × p (pole pair);…- r, q, n, p – with italics

Before Eq. (8a): “ … The n-order Fourier…” – n with italics

etc. etc.

Author Response

Thank you very much for your comment.

We have carefully written our responses to the comments.

Reviewer 2 Report

  1. In Page 3, the unit of TRV value has error, which should be corrected.
  2. The design process is not presented, which does not match the title of the article.
  3. The related references in the introduction part seem inadequate. More references about analytical method should be added, and the merits and drawback of mentioned analytical method should be illustrated.
  4. For the SPM machine, too thin permanent magnet thickness can easily cause irreversible demagnetization. However, too thick permanent magnet thickness would increase the effective air gap length, resulting in a decrease in the output torque of the motor. In 3.4, how to optimize the magnet thickness? It is not scientific to only analyze the amplitude of the first harmonic of the air-gap magnetic density.
  5. The magnetic density produced by PMs and armature at the same time should be added. Besides, the analytical method without considering the slot effect and the saturation effect seems to be of little significance in machine design, because there will be a large error compared to the finite element analysis.
  6. The slot effect has a great influence on the air gap flux density of the motor, resulting in different back-EMF and electromagnetic torque. However, in Fig.12, there is no significant difference in back-EMF between slotless model and slot model. It is better to present the theoretical explanation.

Author Response

(The authors gave the same response as above.)

Reviewer 3 Report

The authors should stress in the manuscript the novelties of their approach, otherwise, their work shows no scientific significance.

Some remarks:

Why was rapid analysis required, as established in point 3.2? It seems that the teeth were not included in the analytical analysis, and saturation there is considerable.

How was saturation considered in the analytical model whose results are portrayed in the first point 3.4?

Why is the displacement current mentioned in the second point 3.4?

How was the number of slots per pole and phase selected?

Author Response

(The authors gave the same response as above.)

Reviewer 4 Report

The following comments and suggestions should be carefully considered and performed to significantly improve the quality of this paper.

1. The section "1. Introduction" is too short and not clearly describe the key necessity and motivation of this research. Furthermore, the last paragraph of section "1. Introduction" has not yet presented the main originality/novelty and salient contributions of the proposed analytical method in this research as compared to existing related works. In the reviewer's opinion, this paper seems as only an application study case with the detailed descriptions of implementation procedure. The authors need to consider and revise these important issues thoroughly.

2. In Fig. 4, why the relative permeability of the PM was set as "u_r = 1"? The authors should explain in more details.

3. In Fig. 5, only the 1st harmonic order in the FFT analysis was used to define the optimal magnet usage point (marked with the star). Is this sufficient and suitable for various operation cases of the PM synchronous generator/motor in real applications?

4. In Fig. 13, the experimental results/tests were only with a load resistance range of [2.4 ohm, 10 ohm], while the load resistance range used in the FEM and proposed analytical method is [1 ohm, 20 ohm]. Therefore, more experimental tests and results are highly recommended to carefully validate the effectiveness of the proposed analytical method.

5. There are some typing issues in mathematical symbols [e.g., between Equation (6) and (7), etc.] and equations [e.g., Equation (13), Equation (14), in Table 2, etc.] in this manuscript. These issue need to be corrected.

6. A section "Nomenclature" to sufficiently and clearly define the main symbols and parameters used in this study should be newly added in the revised paper.

Author Response

(The authors gave the same response as above.)

Round 2

Reviewer 2 Report

Nice reply.

Author Response

Thank you for your interest in our paper.

Reviewer 4 Report

This paper has been revised relatively well. In addition, the following minor comments and suggestions need to be considered and revised thoroughly.

1.  At the line 243 (above Figure 8), the unit for the three currents i_a, i_b, and i_c need to be presented.

2.  At lines 160-161, the indication number of equation (8.a)should be aligned in the same line of formula. At lines 162-163, the indication number of equation (8.b) should be aligned in the same line of formula. At lines 305-306, the indication number of equation (22) should be aligned in the same line of formula. The similar issue at other equations/formulas in whole the manuscript should be checked and revised.

3.  Regarding the lack of experimental results in Figure 13 and Figure 14, the following response of the author: 
"We want more experimental results, but we can't add more due to the lack of number and variety of load resistances. However, as shown in Fig. 13, the accuracy of the characteristic analysis results according to the load resistance is excellent. Therefore, it is considered to have high accuracy of the experimental results at another load resistance point.";
=> These explanations for the above issue should be suitably added and clearly presented in the revised paper.

4. In the paragraph at lines 338-348, the indication number of figures (i.e., Figure 13, and Figure 14) seem be not correctly linked with the contents shown in the figures. Therefore, the indication number of figures in Subsection 4.3 (lines 336-354) need to be checked and revised carefully.

Author Response

Thanks for your hard work.
